# Exploring the research needs, barriers and facilitators to the collection of biological data in adolescence for mental health research: a scoping review protocol paper

Courtney Worrell [ORCID],[1] Rebecca Pollard [ORCID],[1] Tyler Weetman,[2] Zara Sadiq,[2] Maria Pieptan,[1] Gillian Brooks,[3] Matthew Broome,[2,4] Niyah Campbell,[2] Nzinga Gardner,[5] Seeromanie Harding,[6] Anna Lavis,[7] Rosemary R C McEachan,[8] Valeria Mondelli,[1] Craig Morgan,[9,10] Chiara Nosarti,[11,12] Talya Porat,[6,13] David Ryan,[8] Lea Schmid,[1] Katy Shire [ORCID],[8] Anthony Woods,[1] Carmine M Pariante,[1] CELEBRATE Youth Expert Working Group, Paola Dazzan,[1] Rachel Upthegrove[2,4]

For numbered affiliations see end of article.

**Correspondence to**
Courtney Worrell;
courtney.worrell@kcl.ac.uk

## ABSTRACT

**Introduction** While research into adolescent mental health has developed a considerable understanding of environmental and psychosocial risk factors, equivalent biological evidence is lacking and is not representative of economic, social and ethnic diversity in the adolescent population. It is important to understand the possible barriers and facilitators to conduct this research. This will then allow us to improve our understanding of how biology interacts with environmental and psychosocial risk factors during adolescence. The objective of this scoping review is to identify and understand the needs, barriers and facilitators related to the collection of biological data in adolescent mental health research.

**Methods and analysis** Reviewers will conduct a systematic search of PubMed, Medline, Scopus, Cochrane, ERIC, EMBASE, ProQuest, EBSCO Global Health electronic databases, relevant publications and reference lists to identify studies published in the English language at any time. This scoping review will identify published studies exploring mental health/psychopathology outcomes, with biological measures, in participants between the ages of 11 and 18 and examine the reported methodology used for data collection. Data will be summarised in tabular form with narrative synthesis and will use the methodology of Levac *et al*, supplemented by subsequent recommendations from the Joanna Briggs Institute Scoping Review Methodology.

**Ethics and dissemination** Ethical approval is not required for this scoping review. The scoping review will be conducted with input from patient and public involvement, specifically including young people involved in our study ('Co-producing a framework of guiding principles for Engaging representative and diverse cohorts of young peopLE in Biological ReseArch in menTal hEalth'—www.celebrateproject.co.uk) Youth Expert Working Group. Dissemination will include publication in peer-reviewed journals, academic presentations and on the project website.

## STRENGTHS AND LIMITATIONS OF THIS STUDY

⇒ This review is being conducted according to methodological frameworks and guidance such as that reported by Levac *et al* and the Joanna Briggs Institute Scoping Review Methodology.
⇒ The systematic search will include grey literature which reduces publication bias.
⇒ The search will include seven different databases (as well as a manual search of the references) to maximise identification of relevant studies.
⇒ The iterative approach to charting data during extraction ensures that all relevant information will be captured.
⇒ The search is limited to work published in the English language which may exclude some relevant publications.

## INTRODUCTION

Adolescence is a distinct period of significant change. Between the ages of 10 and 19, adolescence sees a string of changes biologically, socially and psychologically. In this important time frame, the development that adolescents undergo has an undeniable impact on longer-term trajectories of health and well-being.[1] Due to these substantial changes, this period provides a window of vulnerability, whereby adolescents may be at increased risk of developing poor mental health.

Mental health problems are a leading cause of health-related disability in children and adolescents, with reports that at least 10% globally experience a mental disorder.[2] The UK Office for National Statistics (2017) suggests that one in eight young people between the

ages of 5 and 19 have experienced at least one form of psychopathology in the UK.[3] In a large UK-based cohort, Cybulski et al have reported an increased incidence in the number of young people seeking support for mental health disorders between 2003 and 2018,[4] suggesting an upward trend, one that has seemingly accelerated in the context of COVID-19. Indeed, following the COVID-19 pandemic, concerns about declining adolescent mental health are now being highlighted in the literature, with a meta-analysis indicating a twofold increase in symptoms of anxiety and depression.[5] It is important to note, however, that approximately half of the studies included in this meta-analysis represent samples from China, and others represent North America and Asia, leading to arguments that they may not represent the global picture, and those from low-income and middle-income countries (LMICs).[6] According to the World Health Organisation (WHO) (2021), depression, anxiety and behavioural disorders, such as attention deficit hyperactivity disorder, are the most common forms of mental disorders experienced by adolescents.[7] Regarding mental disorders in adulthood, the available evidence suggests 75% have onset during adolescence[8] showing clear evidence that adolescence is a critical window, and vulnerability during this time can have a substantial impact on longer-term health, specifically mental health. Poor mental health during this period can additionally have an impact on education, future employment and other longer-term consequences.

Researchers have explored the possible risk factors for mental health disorders in adolescents, focusing on genetic, environmental and psychosocial risk factors. Some of these factors include exposure to childhood trauma,[9][10] bullying[11][12] and difficult family relationships,[13] all of which have been shown to play a role in increasing the risk of developing a mental health problem during adolescence. Other factors include environmental and socioeconomic inequalities.[14] A systematic review including data from 23 countries demonstrated that children and adolescents with socioeconomic disadvantage (based on factors such as household income and poverty), were two to three times more likely to develop mental health difficulties, with a somewhat cumulative effect.[15]

Existing large adolescent cohorts with mental health data tend to lack biological measures. Existing cohorts typically explore only saliva or small-volume blood samples for genotyping and basic biomarkers.[16] So, while this literature begins to demonstrate why adolescents may develop mental health disorders, there is a considerable gap in knowledge from a biological perspective. Specifically, in our understanding of how biology interacts with environmental and psychosocial risk factors to impact mental health during adolescence.[17] Due to the high incidence of mental health disorders occurring during adolescence, it has been argued that it is important to explore biological markers at this early stage for understanding the development of mental health disorders.[18]

Biological evidence, such as fluid-based samples (blood, saliva, urine) and brain imaging can, therefore, be an invaluable addition to research in adolescent mental health as we aim to advance knowledge of biological mechanisms, treatment and stratification.

It is known that adolescents may be resistant to engaging with mental health services, particularly for trauma-related disorders, which has led to assumptions regarding their willingness to become involved in research.[19] Where studies are beginning to explore richer biological measures, recruitment to such studies may encounter several barriers. A school-based pilot study, conducted by Warne et al collecting salivary cortisol samples found that only 11.3% of parents provided consent for their child to participate, despite the coproduction of the protocol with key stakeholders including parents and children,[20] demonstrating some of the possible barriers which can affect this research. When research in child and adolescent populations includes biological data collection, lower recruitment rates may be due to the burden and acceptability of the research; in a study exploring child and adolescent participation in research, Gattuso et al found high rates of refusal to participate, which often related to the use of data collection methods, such as blood sampling, which were perceived to be invasive and burdensome.[21] In a feasibility study, young people were asked to report their subjective experience after completing mock clinical interviews, self-report forms, neuroimaging and blood samples.[22] Most responses indicated that the young people felt safe in all the assessments, and the assessment viewed most favourably was the mock MRI scan, which the authors noted as an important finding given the assumption that in research settings MRI is perceived as burdensome and distressing.[22] The mock interviews and self-report measures, on the other hand, were often reported as stressful and too long.[22] The topic of research being considered 'too sensitive' was another highly reported reason for refusal, which may imply further difficulties when the topic of research surrounds mental ill health.[21] Some studies also demonstrate issues with levels of attrition in such research. Mascarell Maričić et al discuss this issue in the context of the IMAGEN cohort, a longitudinal study of 2000 aged 14 years recruited and followed up to the age of 22. While discussing how attrition can have an impact on longitudinal multimodal statistical modelling, the authors note that they were able to still have complete data for 1300 of the cohort by age 22 which provided this particular study an opportunity to still explore the developmental trajectories in focus.[23] However, a review by Dimolareva et al argued that it is generally difficult to understand more about the nature of recruitment challenges, noncompliance and attrition in research with adolescents as many relevant papers fail to report such information.[24]

Even in the face of good recruitment targets, a lack of representativeness still hinders the generalisability of findings to adolescent populations and potentially those most at risk. For example, Warne et al acknowledged a lack of diversity in

the sample recruited in the salivary cortisol study.[20] Predominantly, research looking at biological mechanisms of mental disorders has focused on adults, and there is a sparsity of biological research on mental disorders, such as depression, in adolescence.[25] Looking at the acceptability of biological data collection in children, Condon highlighted that those who drop out of research or do not participate efficiently, are possibly those who are most important to investigate, tending to be those from under-represented groups,[26] and other research highlights that higher levels of attrition are seen in those with lower socioeconomic status or mental health difficulties.[27] However, it should be noted that this review was again specific to the collection of salivary cortisol data in children and may therefore not be representative of all biological data collection. Where large-scale cohort studies do exist, such as the Adolescent Brain Cognitive Development study, specific efforts have been employed to increase retention in this age group, especially where long-term follow-up occurs.[28] There is also an important gap in the distribution of biological research studies in adolescents between LMICs and high-income countries (HICs). With the exception of projects such as the Identifying Depression Early in Adolescence Risk Stratified Cohort recruiting adolescents in Brazil,[29] most studies in this area are conducted in HIC, despite the fact that 90% of adolescents worldwide live in LMIC.[17 30] The lack of representatives for cohorts of ethnic minorities and vulnerable populations is particularly concerning given the widening inequalities in children's health and mental health, whereby children from deprived areas are facing the worst of the crisis.[31]

Due to these clear barriers to biological research in adolescence and mental health, and the importance of conducting research in this area, it is evident that advancements are needed in the field. While facilitators could be very important to this advancement, there is little dedicated research into possible facilitators for conducting research in this population. Facilitators to adolescents seeking help in a care setting are more widely reported, and mental health literacy, particularly having more information surrounding mental health, was reported as a facilitator.[32] In a research setting, Warne et al note that engaging with stakeholders serves as a facilitator, similarly as it helps to increase understanding of the research and topic area, as well as improving protocols and recruitment.[20]

Given these limitations, and lack of focus on addressing the facilitators for improvement, the Co-producing a framework of guiding principles for Engaging representative and diverse cohorts of young peopLE in Biological ReseArch in menTal hEalth (CELEBRATE) project, co-led in equal partnership by academics and young people, aims to produce a framework of guiding principles as a methodological tool that will operationalise young people's and stakeholders' preferred approach for successfully engaging in biological research in mental health. This work will produce a methodological tool to facilitate biological research in large and representative mental health studies in young people, integrating multiple biological measures, to enhance our ability to draw mechanistic insights into adolescent mental health. Being active decision-makers regarding their own role when participating in research, it is essential for young people to be engaged throughout such work to cocreate the knowledge. A plain language summary, accessible for young people, can be found in the online supplemental material.

It is, therefore, essential to identify the needs of both the population and the research, and the barriers to the research to address and overcome these in future research. There is also value in exploring possible existing, or perceived, facilitators to further aid the development of this important research subject. A preliminary search of PROSPERO, Cochrane Database of Systematic Reviews, Open Science Framework, Joanna Briggs Institute (JBI) Evidence Synthesis and Medline Ovid revealed no current or ongoing scoping or systematic reviews on this topic.

The aim of this scoping review is to gather the current evidence of how to best collect biological data (such as brain imaging and fluid-based samples) from adolescent participants in mental health research, identifying barriers and facilitators affecting research in this field.

### Review question

What are the needs of the population and of the research, the barriers and facilitators, in the collection of biological data in mental health research from adolescent populations?

### METHODS

This review will use the methodological framework of Levac et al,[33] supplemented by subsequent recommendations from the JBI Scoping Review Methodology.[34] The JBI guidance is a further refined set of guidance building on that of Levac et al, specifically focusing on rigorous synthesis. The JBI Manual for Evidence Synthesis[35] has been referred to in the preparation of this work. The design of this work began in early 2023, with the initial search performed in February 2023. The work is scheduled to finish at the end of 2024.

### Search strategy

The search strategy will aim to locate published studies only. Search terms have been identified using the (patient, exposure, outcome) framework which identified the population as young people/adolescents/teenagers/ages 11–18; the exposure as participation in mental health research with biological measurements such as MRI and blood tests; and the outcome as attitudes and beliefs about participation, barriers to recruitment, retention or involvement (see online supplemental appendix I for full search term).

Systematic searches will be undertaken through PubMed, Medline, Scopus, Cochrane, ERIC, EMBASE, ProQuest, EBSCO Global Health electronic databases and manual exploration of relevant publications, as well

as identifying studies through relevant citations. The addition of grey literature aims to reduce the potential influence of publication bias and improve the comprehensiveness of the review, allowing us to identify as much of the relevant literature as possible. The use of grey literature in reviews is important.[36] The search strategy, including all identified keywords and index terms, will undergo minor revision as per the requirements of the databases. Two reviewers will independently conduct the search to ensure that the search is accurate and identifies all relevant publications across databases.

Studies published in the English language will be included and there will not be a time restriction on when they were published.

### Eligibility criteria
► Inclusion criteria:
  – Studies focusing on people aged 11–18 years (studies which recruit participants within this age range will be included even if younger or older participants are also included, eg, child and adolescent samples, and adolescent and young adult samples). Samples which had children up to 11 are not included, and samples which recruited adults who are 18 and above are not included).
  – Research focusing on mental health or psychopathology outcomes and experiences.
  – Research that includes biological data collection methods, including but not limited to, neuroimaging or biological sample collection.
► Exclusion criteria:
  – Papers not written in the English language.
  – Papers without a description of data collection.

### Context
Studies conducted globally will be included in the review. Samples will not be selected based on cultural, ethnicity or sex-based/gender-based factors.

### Types of sources
This scoping review will consider all study designs, including both qualitative and quantitative methodologies. Reviews will be included, and opinion papers will be included if they specifically discuss the collection of biological data in this group.

### Study/source of evidence selection
Following the search, all identified citations will be collated and uploaded into Zotero with duplicates removed. A pilot test will be conducted, with a random subset of 10 papers from the search selected for the independent reviewers to assess against the inclusion criteria, to be checked for consistency. Following the pilot, the titles and abstracts will then be screened by two independent reviewers, with a third reviewer to resolve discrepancies. Reviewers may also screen relevant 'discussion' sections, due to the nature of the research question, as the broader interpretation of results in context may be reported here. Potentially relevant sources will be reviewed in full and assessed against the inclusion and exclusion criteria

by the two reviewers. Reasons for exclusion will be recorded and reported in a Preferred Reporting Items for Systematic Reviews and Meta-analyses extension for Scoping Review flow chart[37] in the scoping review report. Disagreements between the two reviewers will be resolved by the third reviewer.

### Data extraction
Data will be extracted from papers included in the scoping review by reviewers, with an independent reviewer checking the consistency of extraction. A data extraction form will be developed by the reviewers and will include details of the design, including sociodemographics of samples, recruitment and retention, biological measures and outcomes. Key issues to investigate include engagement strategies with young people, partnerships with other key stakeholders, cultural appropriateness of research processes, community engagement, feedback strategies and benefits to schools. Information will be charted on whether strategies were evaluated, appeared to work or not and why. The process of charting the data will be iterative, with reviewers constantly revising the form to ensure that all relevant data is extracted. Modifications will be detailed in the scoping review report. Specific questions may emerge during the review, such as those focusing on culturally appropriate strategies for recruitment, study advocacy in relation to cultural and linguistic diversity, and more.

A draft extraction form is provided in online supplemental appendix II.

### Patient and public involvement
Patient and public involvement (PPI) is at the centre of the CELEBRATE project, of which this scoping review is a part. Our project proposal was developed in consultation with young people and the project as a whole is being co-led by young people. Co-leadership is facilitated through regular meetings with the Youth Expert Working Group (YEWG) as well as the provision of opportunities to provide direct input into project-related activities, such as the approval of search terms specifically relating to this scoping review protocol. Other key stakeholders are also involved in the codesign and governance which includes one of our coauthors for this protocol paper.

Members of our YEWG supported the development of this proposal through a series of meetings in which we discussed search terms and anticipated findings and supported the writing of our plain language summary, found in online supplemental material.

### Data analysis and presentation
Data will be summarised in tabular form with a narrative synthesis. We will perform subgroup analysis and/or reporting as data allows.

### Ethics and dissemination
Ethical approval is not required to conduct this scoping review. The scoping review will be conducted with input from patient and public involvement (PPI), specifically

including young people involved in our Youth Expert Working Group who will provide feedback on the review findings and interpretation.

Dissemination will include publication in peer-reviewed journals, academic presentations and on the project website www.celebrateproject.co.uk. The results of the scoping review will inform other aspects of the wider project, including topic guides for focus groups with stakeholders on the topic area.

**Author affiliations**
[1]Department of Psychological Medicine, Institute of Psychiatry, Psychology, and Neuroscience, King's College London, London, UK
[2]Institute for Mental Health, University of Birmingham, Birmingham, UK
[3]King's Business School, King's College London, London, UK
[4]Birmingham Women's and Children's NHS Foundation Trust, Birmingham, UK
[5]Public Review Member, NIHR, London, UK
[6]Department of Population Health Sciences, Faculty of Life Sciences and Medicine, King's College London, London, UK
[7]Institute of Applied Health Research, University of Birmingham, Birmingham, UK
[8]Bradford Institute for Health Research, Bradford, UK
[9]Health Service and Population Research Department, Institute of Psychiatry, Psychology, and Neuroscience, King's College London, London, UK
[10]ESRC Centre for Society and Mental Health, Institute of Psychiatry, Psychology, and Neuroscience, King's College London, London, UK
[11]Department of Child and Adolescent Psychiatry, Institute of Psychiatry, Psychology and Neuroscience, King's College London, London, UK
[12]Centre for the Developing Brain, Department of Perinatal Imaging and Health, School of Biomedical Engineering and Imaging Sciences, King's College London, London, UK
[13]Dyson School of Design Engineering, Faculty of Engineering, Imperial College London, London, UK

**Acknowledgements** Throughout this project, we have worked with members of the CELEBRATE Youth Expert Working Group (YEWG). This group is formed of young people aged 11–18 with vested interest in youth mental health research. Through online meetings, we engaged YEWG members in conversations and activities that provided us an insight into their views on how we can best conduct this research. Their input was particularly valuable in refining the terms we used to conduct this scoping review. We'd like to thank Abel, Daniel, Fawad, Fera, Kirsten, Lewis, Noa and Tash for all their support.This research is supported by the National Institute for Health Research (NIHR) Mental Health Biomedical Research Centre (BRC) at South London and Maudsley NHS Foundation Trust. This research is supported by the NIHR Oxford Health Biomedical Research Centre.

**Collaborators** CELEBRATE Youth Expert Working Group.

**Contributors** CW has contributed significantly to the drafting and editing of the manuscript. RU, PD, VM and CN are leading this work, conceiving the idea and developing the study methods. SH has advised on methodology. RP, TW, ZS, MP, AW, CW and RU have contributed to the development of the search strategy, which was discussed with all authors, including GB, MB, NC, NG, SH, AL, RM, VM, CM, CN, TP, DR, LS, KS, CP, CELEBRATE YEWG and PD. The CELEBRATE YEWG, RP, PD and NC supported the development of the plain language summary. NC has led and facilitated YEWG meetings that have discussed this work, which the CELEBRATE YEWG, CW, RP, ZS, NG, DR, AW, PD and RU have contributed to. All authors, CW, RP, TW, ZS, MP, GB, MB, NC, NG, SH, AL, RM, VM, CM, CN, TP, DR, LS, KS, AW, CP, CELEBRATE YEWG, PD and RU, have contributed meaningfully to the drafting and editing of the manuscript.

**Funding** This work is being supported by the Medical Research Council (grant number MR/X003078/1).

**Disclaimer** The views expressed are those of the author(s) and not necessarily those of the NIHR or the Department of Health and Social Care.

**Competing interests** None declared.

**Patient and public involvement** Patients and/or the public were involved in the design, or conduct, or reporting, or dissemination plans of this research. Refer to the Methods section for further details.

**Patient consent for publication** Not applicable.

**Provenance and peer review** Not commissioned; externally peer reviewed.

**ORCID iDs**
Courtney Worrell http://orcid.org/0000-0003-0539-2376
Rebecca Pollard http://orcid.org/0000-0002-6516-6891
Katy Shire http://orcid.org/0000-0002-2093-181X

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
