## [Reviewer comments · BMJ Open]

ARTICLE DETAILS

TITLE (PROVISIONAL)	Exploring the research needs, barriers, and facilitators to the collection of biological data in adolescence for mental health research: A Scoping Review Protocol paper
AUTHORS	Worrell, Courtney; Pollard, Rebecca; Weetman, Tyler; Sadiq, Zara; Pieptan, Maria; Brooks, Gillian; Broome, Matthew; Campbell, Niyah; Gardner, Nzinga; Harding, Seeromanie; Lavis, Anna; McEachan, Rosemary; Mondelli, Valeria; Morgan, Craig; Nosarti, Chiara; Porat, Talya; Ryan, David; Schmid, Lea; Shire, Katy; Woods, Anthony; Pariante, Carmine; CELEBRATE Youth Expert Working Group, CELEBRATE Youth Expert Working Group; Dazzan, Paola; Upthegrove, Rachel

VERSION 1 – REVIEW

REVIEWER	Adjorlolo, Samuel University of Ghana
REVIEW RETURNED	20-Nov-2023
GENERAL COMMENTS	This is an important protocol that will significantly advance the field of practice.
REVIEWER	Minnis, Helen University of Glasgow, Mental Health and Wellbeing
REVIEW RETURNED	04-Dec-2023
GENERAL COMMENTS	This is a well-justified and clear account of a
REVIEWER	Ruch, Donna A Abigail Wexner Research Institute at Nationwide Children's Hospital
REVIEW RETURNED	17-Feb-2024
GENERAL COMMENTS	Unclear why the authors chose an age range 11-18 years - why not full adolescence between the ages of 10-19? and why specific grey literature? Would also provide more information around what is meant by "biological" evidence and why this knowledge/research is important to the field. Also, a bit more clarity around the Joanna Briggs Institute Scoping Review Methodology would be helpful.

VERSION 1 – AUTHOR RESPONSE

Reviewer 1:
Dr. Samuel Adjorlolo, University of Ghana

This is an important protocol that will significantly advance the field of practice.

- Thank you so much for your kind comment.

Reviewer 2:

Prof. Helen Minnis, University of Glasgow

This is a well-justified and clear account of a

- We would like to thank you for the kind words regarding this work.

Reviewer 3:

Dr. Donna A Ruch, Abigail Wexner Research Institute at Nationwide Children's Hospital

Unclear why the authors chose an age range 11-18 years – why not full adolescence between the ages of 10-19?

- Thank you for your comment. We have chosen to explore the age range of 11-18 as this is the first stage of a larger project being conducted in the UK. We are consistently looking at adolescents within the secondary school age which is 11-18 years in the UK. In addition, as we expected a high volume of papers to be identified in the review due to the wide scope, it was decided that keeping the papers to this age would also be beneficial due to heterogeneity when looking at the recruitment of younger samples in primary school ages. We are also aware that choosing papers which strictly recruited this age range would be limiting and so papers which have recruited participants within this age range have been recruited regardless of whether they also included younger or older participants meaning that we have tried to capture any studies which recruited adolescents. In order to make this clear in our protocol paper, we have expanded on this point of the eligibility criteria to state that “(studies which recruit participants within this age range will be included even if younger or older participants are also included, for example, child and adolescent samples, and adolescent and young adult samples). Samples which had children up to 11 are not included and samples which recruited adults who are 18 and above are not included).”

and why specific grey literature?

- We have decided to include grey literature in order to reduce the risk of publication bias in our review and to improve the comprehensiveness of our search. To make this inclusion clearer in our paper we have expanded on this point in the second paragraph of our search strategy: “The addition of grey literature aims to reduce the potential influence of publication bias and improve the comprehensiveness of the review, allowing us to identify as much of the relevant literature as possible. The use of grey literature in reviews is important [36]”

Would also provide more information around what is meant by “biological” evidence and why this knowledge/research is important to the field.

- Thank you very much for this important point. In the introduction (specifically the 4th paragraph) where the lack of biological measures in this body of research is introduced, we have added a sentence to demonstrate the importance of such evidence “Due to the high incidence of mental health disorders occurring during adolescence, it has been argued that it is important to explore biological markers at this early stage for understanding the development of mental health disorders [18]. Biological evidence, such as fluid-based samples (blood, saliva, urine) and brain imaging can therefore be an invaluable addition to research in adolescent mental health as we aim to advance knowledge of biological mechanisms, treatment and stratification” In addition. The last sentence of the introduction proposes the aim of the scoping review and in this, we have now included in brackets, examples of biological data, “The aim of this scoping review is to gather the current evidence of how to best collect biological data (such as brain imaging and fluid-based samples) from adolescent participants in mental health research, identifying barriers and facilitators affecting research in this field.”

Also, a bit more clarity around the Joanna Briggs Institute Scoping Review Methodology would be helpful.

- Thank you for highlighting this. The Joanna Briggs Institute Scoping Review Methodology has been

a very helpful resource for the conduct of this work. We realise that we did not previously provide much information about this and so have now added to the first paragraph of the 'Methods' section, that the guidance builds upon that of Levac et al., (2010) and that the evidence synthesis manual was referred to during the conduct. We have additionally cited this manual more widely than the chapter specific to the scoping review methodology. We hope that this provides some more clarity surrounding the methodology. This addition reads, "The JBI guidance is a further refined set of guidance building on that of Levac et al., specifically focussing on rigorous synthesis. The JBI Manual for Evidence Synthesis [35] has been referred to in the preparation of this work."